# Real-time Photorealistic Dynamic Scene Representation and Rendering with 4D Gaussian Splatting

**Zeyu Yang, Hongye Yang, Zijie Pan, Li Zhang**[*]
Fudan University

https://fudan-zvg.github.io/4d-gaussian-splatting

## Abstract

Reconstructing dynamic 3D scenes from 2D images and generating diverse views over time is challenging due to scene complexity and temporal dynamics. Despite advancements in neural implicit models, limitations persist: (i) *Inadequate Scene Structure*: Existing methods struggle to reveal the spatial and temporal structure of dynamic scenes from directly learning the complex 6D plenoptic function. (ii) *Scaling Deformation Modeling*: Explicitly modeling scene element deformation becomes impractical for complex dynamics. To address these issues, we consider the spacetime as an entirety and propose to approximate the underlying spatio-temporal 4D volume of a dynamic scene by optimizing a collection of 4D primitives, with explicit geometry and appearance modeling. Learning to optimize the 4D primitives enables us to synthesize novel views at any desired time with our tailored rendering routine. Our model is conceptually simple, consisting of a 4D Gaussian parameterized by anisotropic ellipses that can rotate arbitrarily in space and time, as well as view-dependent and time-evolved appearance represented by the coefficient of 4D spherindrical harmonics. This approach offers simplicity, flexibility for variable-length video and end-to-end training, and efficient real-time rendering, making it suitable for capturing complex dynamic scene motions. Experiments across various benchmarks, including monocular and multi-view scenarios, demonstrate our 4DGS model's superior visual quality and efficiency.

## 1 Introduction

Modeling dynamic scenes from 2D images and rendering photorealistic novel views in real-time is crucial in computer vision and graphics. This task has been receiving increasing attention from both industry and academia because of its potential value in a wide range of AR/VR applications. Recent breakthroughs, such as NeRF (Mildenhall et al., 2020), have achieved photorealistic static scene rendering (Barron et al., 2021; Verbin et al., 2022). However, adapting these techniques to dynamic scenes is challenging due to several factors. Object motion complicates reconstruction, and temporal scene dynamics add significant complexity. Moreover, real-world applications often capture dynamic scenes as monocular videos, making it impractical to train separate static scene representations for each frame and then combine them into a dynamic scene model. The central challenge is preserving intrinsic correlations and sharing relevant information across different time steps while minimizing interference between unrelated spacetime locations.

Dynamic novel view synthesis methods can be categorized into two groups. The first group employs structures such as MLPs (Li et al., 2022b) or grids (Wang et al., 2023), including their low-rank decompositions (Fridovich-Keil et al., 2023; Cao & Johnson, 2023; Attal et al., 2023), to learn a 6D plenoptic function without explicitly modeling scene motion. The effectiveness of these methods in capturing correlations across different spatial-temporal locations depends on the inherent characteristics of the chosen data structure. However, they lack flexibility in adapting to the underlying scene motion. Consequently, these methods either suffer from parameter sharing across spatial-temporal

---

[*]Li Zhang (lizhangfd@fudan.edu.cn) is the corresponding author with School of Data Science, Fudan University.

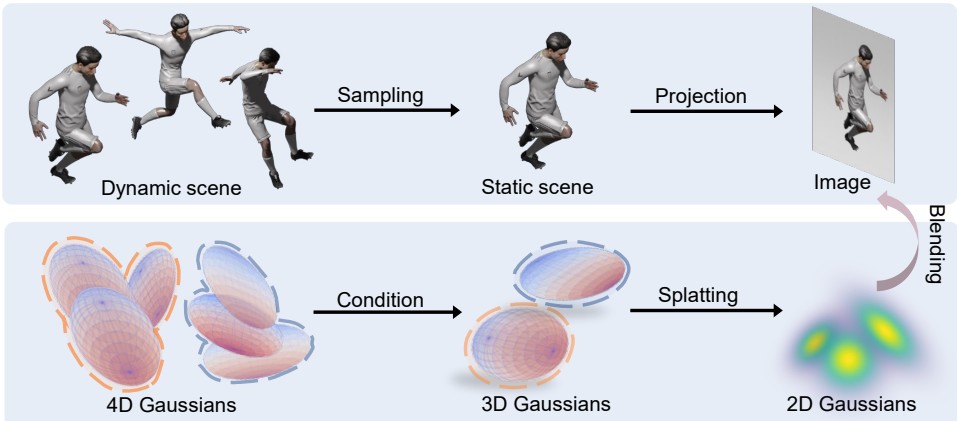

Figure 1: **Schematic illustration of the proposed 4DGS.** This diagram illustrates why our 4D primitive is naturally suitable for representing dynamic scenes. Within our framework, the transition from 4D Gaussian to 2D Planar Gaussian can conceptually correspond to the process where a dynamic scene transforms into a 2D image through a temporal orthogonal projection and a spatial perspective projection.

locations, leading to potential interference, or they operate too independently, struggling to harness the inherent correlations resulting from object motion.

In contrast, another group suggests that scene dynamics are induced by the motion or deformation of a consistent underlying representation (Pumarola et al., 2020; Song et al., 2023; Abou-Chakra et al., 2022; Luiten et al., 2024). These methods explicitly learn scene motion, providing the potential for better utilization of correlations across space and time. Nevertheless, they exhibit reduced flexibility and scalability in complex real-world scenes compared to the first group of methods.

To overcome these limitations, this study reformulates the task by approximating a scene's underlying spatio-temporal 4D volume by a set of 4D Gaussians. Notably, 4D rotations enable the Gaussian to fit the 4D manifold and capture scene intrinsic motion. Furthermore, we introduce Spherindrical Harmonics as a generalization of Spherical Harmonics for dynamic scenes to model time evolution of appearance in dynamic scenes. This approach marks the first-ever model supporting end-to-end training and real-time rendering of high-resolution, photorealistic novel views in complex dynamic scenes with volumetric effects and varying lighting conditions. Additionally, our proposed representation is interpretable, highly scalable, and adaptable in both spatial and temporal dimensions.

Our contributions are as follows: **(i)** We propose coherent integrated modeling of the space and time dimensions for dynamic scenes by formulating unbiased 4D Gaussian primitives along with a dedicated splatting-based rendering pipeline. **(ii)** The 4D Spherindrical Harmonics of our method is useful and interpretable to model the time evolution of view-dependent color in dynamic scenes. **(iii)** Extensive experiments on various datasets, including synthetic and real, monocular, and multi-view, demonstrate that our method outperforms all previous methods in terms of visual quality and efficiency. Notably, our method can produce photo-realistic, high-resolution video at speeds far beyond real-time.

## 2 RELATED WORK

**Novel view synthesis for static scenes** In recent years, the field of novel view synthesis has received widespread attention. Mildenhall et al. (2020) is the pioneering work that initiated this trend, suggesting using an MLP to learn the radiance field and employing volumetric rendering to synthesize images for any viewpoint. However, vanilla NeRF requires querying the MLP for hundreds of points each ray, significantly constraining its training and rendering speed. Some subsequent works have attempted to improve the speed, such as employing well-tailored data structures (Chen et al., 2022; Sun et al., 2022; Hu et al., 2022; Chen et al., 2023), discarding large MLP (Fridovich-Keil et al., 2022), or adopting hash encodings (Müller et al., 2022). Other works (Zhang et al., 2020; Verbin et al., 2022; Barron et al., 2021; 2022; 2023) aim to enhance rendering quality by addressing

existing issues in the vanilla NeRF, such as aliasing and reflection. Yet, these methods remain confined to the nuances of differentiable volume rendering. In contrast, Kerbl et al. (2023) introduced 3D Gaussian Splatting, a novel framework that possesses the advantages of volumetric rendering approaches—offering high-fidelity view synthesis for complex scenes, while also benefiting from the merits of rasterization approaches, enabling real-time rendering for large-scale scenes. Inspired by this work, in this paper, we further demonstrate that Gaussian primitives are also an excellent representation of dynamic scenes.

**Novel view synthesis for dynamic scenes** Synthesizing novel views of a dynamic scene at a desired time from a series of 2D captures is a more challenging task. The intricacy lies in capturing the intrinsic correlation across different timesteps. This task cannot be trivially regarded as an accumulation of novel view synthesis for the static scene of each frame, as such an approach is prohibitively expensive, scales poorly for synthesizing new views at a time beyond training data, and inevitably fails when observations at a single frame are insufficient to reconstruct the entire scene. Inspired by the success of NeRF, one research line attempts to learn a 6D plenoptic function represented by a well-tailored implicit or explicit structure without direct modeling for the underlying motion to address this challenge (Li et al., 2022b; Fridovich-Keil et al., 2023; Cao & Johnson, 2023; Wang et al., 2023; Attal et al., 2023). However, these methods struggle with the coupling between parameters. An alternative approach explicitly models continuous motion or deformation, presuming the dynamics of the scene result from the movement or deformation of particular static structures, like particle or canonical fields (Pumarola et al., 2020; Song et al., 2023; Abou-Chakra et al., 2022; Luiten et al., 2024). Among them, point-based approaches have consistently been deemed promising. Recently, drawing inspiration from 3D Gaussian Splatting, Luiten et al. (2024) represented dynamic scenes with a set of simplified 3D Gaussians shared across timesteps and optimized them frame-by-frame. With the physically-based priors encoded in its regularizations, dynamic 3D Gaussians can be optimized to represent dynamic scenes faithfully given its multi-view captures, and achieve long-term tracking by exploiting dense correspondences across timesteps.

**Dynamic 3D Gaussians** Recently, there have been substantial efforts (Chen & Wang, 2024) in extending 3D Gaussian Splatting into dynamic scenes. Beyond the pioneer work Luiten et al. (2024) mentioned above, Yang et al. (2023); Wu et al. (2023); Liang et al. (2023) propose to model the geometry and dynamic of scenes by the joint optimization of Gaussians in canonical space and a deformation field. (Kratimenos et al., 2023) encourages locality and rigidity between points by factorizing the motion in the scene into a few neural trajectories. These works ingeniously incorporate a prior of topological invariance into their representations, making them well-suited for reconstructing dynamic scenes from monocular videos. However, they assume that dynamic scenes are generated by a fixed set of 3D Gaussians and the elements composing the scene are always visible. In contrast, by formulating a novel 4D scene primitive, we discard their underlying assumptions and circumvent the need to maintain ambiguous and complex tracking relationships, thereby facilitating a more flexible and versatile approach to handling complex scenes in real-world applications.

## 3 METHOD

We propose a novel photorealistic scene representation tailored for modeling general dynamic scenes. In this section, we will delineate each component of it and the corresponding optimization process. In Section 3.1, we will begin by reviewing 3D Gaussian Splatting (Kerbl et al., 2023) from which our method inspired. In Section 3.2, we detail how our 4D Gaussian represents dynamic scenes and synthesizes novel views. An overview is shown in Figure 2. The optimization framework will be introduced in Section 3.3.

### 3.1 PRELIMINARY: 3D GAUSSIAN SLATTING

3D Gaussian Splatting (Kerbl et al., 2023) employs anisotropic Gaussian to represent static 3D scenes. Facilitated by a well-tailored GPU-friendly rasterizer, this representation enables real-time synthesis of high-fidelity novel views.

**Representation of 3D Gaussians** In 3D Gaussian Splatting, a scene is represented as a cloud of 3D Gaussians. Each Gaussian has a theoretically infinite scope and its influence on a given spatial

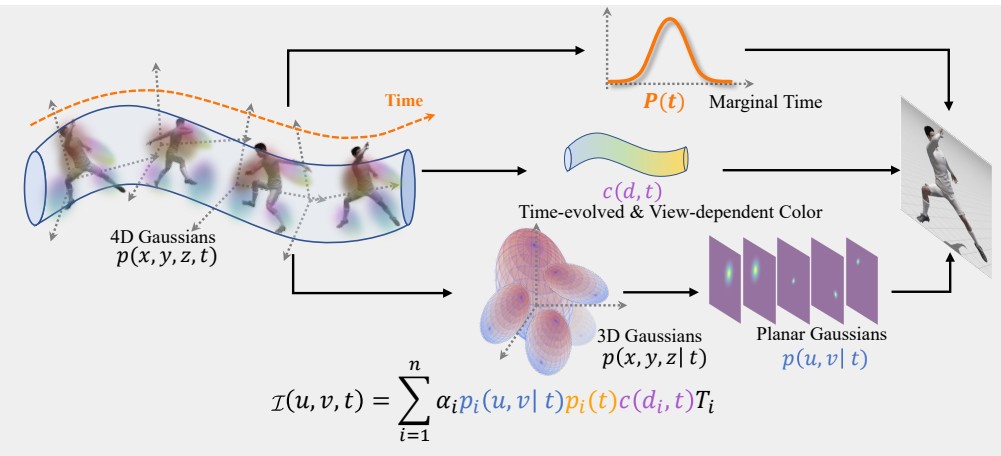

$$\mathcal{I}(u, v, t) = \sum_{i=1}^{n} \alpha_i p_i(u, v \mid t) p_i(t) c(d_i, t) T_i$$

Figure 2: **Rendering pipeline of our 4DGS.** Given a time $t$ and view $\mathcal{I}$, each 4D Gaussian is first decomposed into a conditional 3D Gaussian and a marginal 1D Gaussian. Subsequently, the conditional 3D Gaussian is projected to a 2D splat. Finally, we integrate the planar conditional Gaussian, 1D marginal Gaussian, and time-evolving view-dependent color to render the view $\mathcal{I}$.

position $x \in \mathbb{R}^3$ defined by an unnormalized Gaussian function:

$$p(x|\mu, \Sigma) = e^{-\frac{1}{2}(x-\mu)^T \Sigma^{-1}(x-\mu)}, \tag{1}$$

where $\mu \in \mathbb{R}^3$ is its mean vector, and $\Sigma \in \mathbb{R}^{3 \times 3}$ is an anisotropic covariance matrix. In the Appendix, we will show that it holds desired properties of normalized Gaussian probability density function critical for our methodology, i.e., the unnormalized Gaussian function of a multivariate Gaussian can be factorized as the production of the unnormalized Gaussian functions of its condition and margin distributions. Hence, for brevity and without causing misconceptions, we do not specifically distinguish between equation 1 and its normalized version in subsequent sections.

In Kerbl et al. (2023), the mean vector $\mu$ of a 3D Gaussian is parameterized as $\mu = (\mu_x, \mu_y, \mu_z)$, and the covariance matrix $\Sigma$ is factorized into a scaling matrix $S$ and a rotation matrix $R$ as $\Sigma = RSS^T R^T$. Here $S$ is summarized by its diagonal elements $S = \text{diag}(s_x, s_y, s_z)$, whilst $R$ is constructed from a unit quaternion $q$. Moreover, a 3D Gaussian also includes a set of coefficients of spherical harmonics (SH) for representing view-dependent color, along with an opacity $\alpha$.

All of the above parameters can be optimized under the supervision of the rendering loss. During the optimization process, 3D Gaussian Splatting also periodically performs densification and pruning on the collection of Gaussians to further improve the geometry and the rendering quality.

**Differentiable rasterization via Gaussian splatting** In rendering, given a pixel $(u, v)$ in view $\mathcal{I}$ with extrinsic matrix $E$ and intrinsic matrix $K$, its color $\mathcal{I}(u, v)$ can be computed by blending visible 3D Gaussians that have been sorted according to their depth, as described below:

$$\mathcal{I}(u, v) = \sum_{i=1}^{N} p_i(u, v; \mu_i^{2d}, \Sigma_i^{2d}) \alpha_i c_i(d_i) \prod_{j=1}^{i-1} (1 - p_i(u, v; \mu_i^{2d}, \Sigma_i^{2d}) \alpha_j), \tag{2}$$

where $c_i$ denotes the color of the $i$-th visible Gaussian from the viewing direction $d_i$, $\alpha_i$ represents its opacity, and $p_i(u, v)$ is the probability density of the $i$-th Gaussian at pixel $(u, v)$.

To compute $p_i(u, v)$ in the image space, we linearize the perspective transformation as in Zwicker et al. (2002); Kerbl et al. (2023). Then, the projected 3D Gaussian can be approximated by a 2D Gaussian. The mean of the derived 2D Gaussian is obtained as:

$$\mu_i^{2d} = \text{Proj}(\mu_i | E, K)_{1:2}, \tag{3}$$

where $\text{Proj}(\cdot | E, K)$ denotes the transformation from the world space to the image space given the intrinsic $K$ and extrinsic $E$. The covariance matrix is given by

$$\Sigma_i^{2d} = (JE\Sigma E^T J^T)_{1:2,1:2}, \tag{4}$$

where $J$ is the Jacobian matrix of the perspective projection.

## 3.2 4D GAUSSIAN FOR DYNAMIC SCENES

**Problem formulation and 4D Gaussian splatting** To extend the formulation of Kerbl et al. (2023) for modeling dynamic scenes, reformulation is necessary. In dynamic scenes, a pixel under view $\mathcal{I}$ can no longer be indexed solely by a pair of spatial coordinates $(u, v)$ in the image plane; But an additional timestamp $t$ comes into play and intervenes. Formally this is formulated by extending equation 2 as:

$$\mathcal{I}(u, v, t) = \sum_{i=1}^{N} p_i(u, v, t) \alpha_i c_i(d) \prod_{j=1}^{i-1} (1 - p_j(u, v, t) \alpha_j). \tag{5}$$

Note that $p_i(u, v, t)$ can be further factorized as a product of a conditional probability $p_i(u, v|t)$ and a marginal probability $p_i(t)$ at time $t$, yielding:

$$\mathcal{I}(u, v, t) = \sum_{i=1}^{N} p_i(t) p_i(u, v|t) \alpha_i c_i(d) \prod_{j=1}^{i-1} (1 - p_j(t) p_j(u, v|t) \alpha_j). \tag{6}$$

Let the underlying $p_i(x, y, z, t)$ be a 4D Gaussian. As the conditional distribution $p(x, y, z|t)$ is also a 3D Gaussian, we can similarly derive $p(u, v|t)$ as a planar Gaussian whose mean and covariance matrix are parameterized by equation 3 and equation 4, respectively.

Subsequently, it comes to the question of *how to represent a 4D Gaussian*. A natural solution is that we adopt a distinct perspective for space and time, that is, considering $(x, y, z)$ and $t$ are independent of each other, i.e., $p_i(x, y, z|t) = p_i(x, y, z)$. Under this assumption, equation 6 can be implemented by adding an extra 1D Gaussian $p_i(t)$ into the original 3D Gaussians (Kerbl et al., 2023). This design can be viewed as imbuing a 3D Gaussian with temporal extension, or weighting down its opacity when the rendering timestep is away from the expectation of $p_i(t)$. However, we show later on that this approach can achieve a reasonable fitting of 4D manifold but is difficult to capture the underlying motion of the scene (see "No-4DRot" in Table 3).

**Representation of 4D Gaussian** To address the mentioned challenge, we suggest to treat time and space dimensions equally by formulating a coherent integrated 4D Gaussian model. Similar to Kerbl et al. (2023), we parameterize its covariance matrix $\Sigma$ as the configuration of a 4D ellipsoid for easing model optimization:

$$\Sigma = RSS^T R^T, \tag{7}$$

where $S$ is a scaling matrix and $R$ is a 4D rotation matrix. Since $S$ is diagonal, it can be completely inscribed by its diagonal elements as $S = \mathrm{diag}(s_x, s_y, s_z, s_t)$. On the other hand, a rotation in 4D Euclidean space can be decomposed into a pair of isotropic rotations, each of which can be represented by a quaternion.

Specifically, given $q_l = (a, b, c, d)$ and $q_r = (p, q, r, s)$ denoting the left and right isotropic rotations respectively, $R$ can be constructed by:

$$R = L(q_l) R(q_r) = \begin{pmatrix} a & -b & -c & -d \\ b & a & -d & c \\ c & d & a & -b \\ d & -c & b & a \end{pmatrix} \begin{pmatrix} p & -q & -r & -s \\ q & p & s & -r \\ r & -s & p & q \\ s & r & -q & p \end{pmatrix}. \tag{8}$$

The mean of a 4D Gaussian can be represented by four scalars as $\mu = (\mu_x, \mu_y, \mu_z, \mu_t)$. Thus far we arrive at a complete representation of the general 4D Gaussian.

Subsequently, the conditional 3D Gaussian can be derived from the properties of the multivariate Gaussian with:

$$\begin{aligned} \mu_{xyz|t} &= \mu_{1:3} + \Sigma_{1:3,4} \Sigma_{4,4}^{-1} (t - \mu_t), \\ \Sigma_{xyz|t} &= \Sigma_{1:3,1:3} - \Sigma_{1:3,4} \Sigma_{4,4}^{-1} \Sigma_{4,1:3} \end{aligned} \tag{9}$$

Since $p_i(x, y, z|t)$ is a 3D Gaussian, $p_i(u, v|t)$ in equation 6 can be derived in the same way as in equation 3 and equation 4. Moreover, the marginal $p_i(t)$ is also a Gaussian in one-dimension:

$$p(t) = \mathcal{N}(t; \mu_4, \Sigma_{4,4}) \tag{10}$$

So far we have a comprehensive implementation of equation 6. Subsequently, we can adapt the highly efficient tile-based rasterizer proposed in Kerbl et al. (2023) to approximate this process, through considering the marginal distribution $p_i(t)$ when accumulating colors and opacities.

**4D spherindrical harmonics**  The view-dependent color $c_i(d)$ in equation 6 is represented by a series of SH coefficients in the original 3D Gaussian Splatting (Kerbl et al., 2023). To more faithfully model the dynamic scenes of real world, we must enable appearance variation with varying viewpoints and also their colors to evolve over time.

Leveraging the flexibility of our framework, a straightforward solution is to directly use different Gaussians to represent the same point at different times. However, this approach leads to duplicated and redundant representation of identical objects, making it challenging to optimize. Instead we choose to exploit 4D extension of the spherical harmonics (SH) that directly represents the time evolution of appearance of each Gaussian. The color in equation 6 could then be manipulated with $c_i(d, t)$, where $d = (\theta, \phi)$ is the normalized view direction under spherical coordinates and $t$ is the time difference between the expectation of the given Gaussian and the viewpoint.

Inspired by studies on head-related transfer function, we propose to represent $c_i(d, t)$ as the combination of a series of 4D spherindrical harmonics (4DSH) which are constructed by merging SH with different 1D-basis functions. For computational convenience, we use the Fourier series as the adopted 1D-basis functions. Consequently, 4DSH can be expressed as:

$$Z_{nl}^m(t, \theta, \phi) = \cos\left(\frac{2\pi n}{T}t\right) Y_l^m(\theta, \phi), \tag{11}$$

where $Y_l^m$ is the 3D spherical harmonics. The index $l \geq 0$ denotes its degree, and $m$ is the order satisfying $-l \leq m \leq l$. The index $n$ is the order of the Fourier series. The 4D spherindrical harmonics form an orthonormal basis in the spherindrical coordinate system.

### 3.3 Training

Following 3D Gaussian Splatting (Kerbl et al., 2023), we conduct interleaved optimization and density control during training. It is worth highlighting that our optimization process is entirely end-to-end, capable of processing entire videos, with the ability to sample at any time and view, as opposed to the traditional frame-by-frame or multi-stage training approaches.

**Optimization**  In optimization, we only use the rendering loss as supervision. In most scenes, combining the representation introduced above with the default training schedule as in Kerbl et al. (2023) is sufficient to yield satisfactory results. However, in some scenes with more dramatic changes, we observe issues such as temporal flickering and jitter. We consider this may arise from suboptimal sampling techniques. Rather than adopting the prior regularization, we discover that straightforward batch sampling in time turns out to be superior, resulting in more seamless and visually pleasing appearance of dynamic visual contents.

**Densification in spacetime**  In terms of density control, simply considering the average magnitude of view space position gradients is insufficient to assess under-reconstruction and over-reconstruction over time. To address this, we incorporate the average gradients of $\mu_t$ as an additional density control indicator. Furthermore, in regions prone to over-reconstruction, we employ joint spatial and temporal position sampling during Gaussian splitting.

## 4 Experiments

In this section, we present comparisons with state-of-the-art methods on two well-established datasets for dynamic scene novel view synthesis: Plenoptic Video dataset (Li et al., 2022b) and D-NeRF dataset (Pumarola et al., 2020). Additionally, we perform ablation studies to gain insights into our method and illustrate the efficacy of key design decisions. Video results and more use cases can be found in our supplementary material.

### 4.1 Experimental setup

#### 4.1.1 Datasets

**Plenoptic Video dataset (Li et al., 2022b)** comprises six real-world scenes, each lasting ten seconds. For each scene, one view is reserved for testing while other views are used for training. To

Table 1: **Comparison with the state-of-the-art methods on the Plenoptic Video benchmark.** [1]: Only report the result on the scene *flames salmon*. [2]: Only report SSIM instead of MS-SSIM like others. [3]: Measured by ourselves using their official released code. [4]: Results on *Spinach*, *Beef*, and *Steak scenes*.

| Method | PSNR ↑ | DSSIM ↓ | LPIPS ↓ | FPS ↑ |
|---|---|---|---|---|
| *- Plenoptic Video (real, multi-view)* | | | | |
| Neural Volumes (Lombardi et al., 2019)[1] | 22.80 | 0.062 | 0.295 | - |
| LLFF (Mildenhall et al., 2019)[1] | 23.24 | 0.076 | 0.235 | - |
| DyNeRF (Li et al., 2022b)[1] | 29.58 | 0.020 | 0.099 | 0.015 |
| HexPlane (Cao & Johnson, 2023) | 31.70 | 0.014 | 0.075 | 0.56[3] |
| K-Planes-explicit (Fridovich-Keil et al., 2023) | 30.88 | 0.020 | - | 0.23[3] |
| K-Planes-hybrid (Fridovich-Keil et al., 2023) | 31.63 | 0.018 | - | - |
| MixVoxels-L (Wang et al., 2023) | 30.80 | 0.020 | 0.126 | 16.7 |
| StreamRF (Li et al., 2022a)[1] | 29.58 | - | - | 8.3 |
| NeRFPlayer (Song et al., 2023) | 30.69 | 0.035[2] | 0.111 | 0.045 |
| HyperReel (Attal et al., 2023) | 31.10 | 0.037[2] | 0.096 | 2.00 |
| 4DGS (Wu et al., 2023)[4] | 31.02 | 0.030 | 0.150 | 36 |
| **4DGS (Ours)** | **32.01** | **0.014** | **0.055** | **114** |

initialize the Guassians for this dataset, we utilize the colored point cloud generated by COLMAP from the first frame of each scene. The timestamps of each point are uniformly distributed across the scene's duration.

**D-NeRF dataset (Pumarola et al., 2020)** is a monocular video dataset comprising eight videos of synthetic scenes. Notably, during each time step, only a single training image from a specific viewpoint is accessible. To assess model performance, we employ standard test views that originate from novel camera positions not encountered during the training process. These test views are taken within the time range covered by the training video. In this dataset, we utilize 100,000 randomly selected points, evenly distributed within the cubic volume defined by $[-1.2, 1.2]^3$, and set their initial mean as the scene's time duration.

## 4.2 IMPLEMENTATION DETAILS

To assess the versatility of our approach, we did not extensively fine-tune the training schedule across different datasets. By default, we conducted training with a total of 30,000 iterations, a batch size of 4, and halted densification at the midpoint of the schedule. We adopted the settings of Kerbl et al. (2023) for hyperparameters such as loss weight, learning rate, and threshold. At the outset of training, we initialized both $q_l$ and $q_r$ as unit quaternions to establish identity rotations and set the initial time scaling to half of the scene's duration. While the 4D Gaussian theoretically extends infinitely, we applied a Gaussian filter with marginal $p(t) < 0.05$ when rendering the view at time $t$. For scenes in the Plenoptic Video dataset, we further initialized the Gaussian with 100,000 extra points distributed uniformly on the sphere encompassing the entire scene to fit the distant background that colmap failed to reconstruct and terminate its optimization after 10,000 iterations. Following the previous work, the LPIPS (Zhang et al., 2018) in the Plenoptic Video dataset and the D-NeRF dataset are computed using AlexNet (Krizhevsky et al., 2012) and VGGNet (Simonyan & Zisserman, 2014) respectively.

## 4.3 RESULTS OF DYNAMIC NOVEL VIEW SYNTHESIS

**Results on the multi-view real scenes** Table 1 presents a quantitative evaluation on the Plenoptic Video dataset. Our approach not only significantly surpasses previous methods in terms of rendering quality but also achieves substantial speed improvements. Notably, it stands out as the sole method capable of real-time rendering while delivering high-quality dynamic novel view synthesis within this benchmark. To complement this quantitative assessment, we also offer qualitative comparisons on the "flame salmon" scene, as illustrated in Figure 3. The quality of synthesis in dynamic regions notably excels when compared to other methods. Several intricate details, including the black bars

Table 2: **Qualitative comparison on monocular dynamic scenes.** The results are averaged over all scenes in the D-NeRF dataset. [1]: rendering at 800×800, otherwise downsampled 2x by default.

| Method | PSNR ↑ | SSIM ↑ | LPIPS ↓ |
|---|---|---|---|
| - *D-NeRF (synthetic, monocular)* | | | |
| T-NeRF (Pumarola et al., 2021) | 29.51 | 0.95 | 0.08 |
| D-NeRF (Pumarola et al., 2021) | 29.67 | 0.95 | 0.07 |
| TiNeuVox (Fang et al., 2022) | 32.67 | 0.97 | 0.04 |
| HexPlanes (Cao & Johnson, 2023) | 31.04 | 0.97 | 0.04 |
| K-Planes-explicit (Fridovich-Keil et al., 2023) | 31.05 | 0.97 | - |
| K-Planes-hybrid (Fridovich-Keil et al., 2023) | 31.61 | 0.97 | - |
| V4D (Gan et al., 2023) | 33.72 | 0.98 | 0.02 |
| 4DGS (Wu et al., 2023)[1] | 33.30 | 0.98 | 0.03 |
| **4DGS (Ours)** | **34.09** | **0.98** | **0.02** |

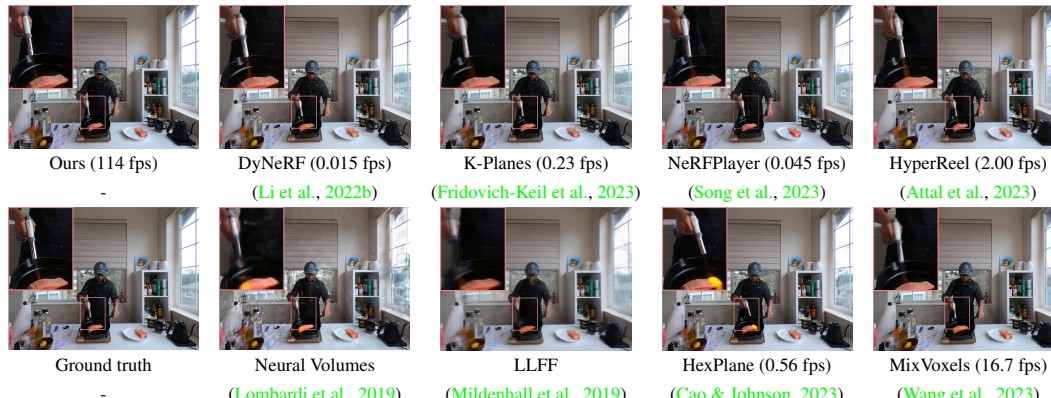

Figure 3: Qualitative result on the *flame salmon*. It can be clearly seen that the visual quality is higher than other methods in the region from the moving hands and flame gun to the static salmon.

on the flame gun, the fine features of the right-hand fingers, and the texture of the salmon, are faithfully reconstructed, demonstrating the strength of our approach.

**Results on the monocular synthetic videos** We also evaluate our approach on monocular dynamic scenes, a task known for its inherent complexities. Previous successful methods often rely on architectural priors to handle the underlying topology, but we refrain from introducing such assumptions when applying our 4D Gaussian model to monocular videos. Remarkably, our method surpasses all competing methods, as illustrated in Table 2. This outcome underscores the ability of our 4D Gaussian model to efficiently exchange information across various time steps.

## 4.4 ABLATION AND ANALYSIS

**Coherent comprehensive 4D Gaussian** Our novel approach involves treating 4D Gaussian distributions without strict separation of temporal and spatial elements. In Section 3.2, we discussed an intuitive method to extend 3D Gaussians to 4D Gaussians, as expressed in equation 6. This method assumes independence between the spatial $(x, y, z)$ and temporal variable $t$, resulting in a block diagonal covariance matrix. The first three rows and columns of the covariance matrix can be processed similarly to 3D Gaussian splatting. We further additionally incorporate 1D Gaussian to account for the time dimension.

To compare our unconstrained 4D Gaussian with this baseline, we conduct experiments on two representative scenes, as shown in Table 3. We can observe the clear superiority of our unconstrained 4D Gaussian over the constrained baseline. This underscores the significance of our unbiased and coherent treatment of both space and time aspects in dynamic scenes.

**4D Gaussian is capable of capturing the underlying 3D movement** Incorporation of 4D rotation to our 4D Gaussian equips it with the ability to model the motion. Note that 4D rotation can result in

Table 3: **Ablation studies.** We ablate our framework on two representative real scenes, *flame salmon* and *cut roasted beef*, which have volumetric effects, non-Lambertian surfaces, and different lighting conditions. "No-4DRot" denotes restricting the space and time independent of each other.

| | Flame Salmon | | Cut Roasted Beef | | Average | |
|---|---|---|---|---|---|---|
| | PSNR ↑ | SSIM ↑ | PSNR ↑ | SSIM ↑ | PSNR ↑ | SSIM ↑ |
| No-4DRot | 28.78 | 0.95 | 32.81 | 0.971 | 30.79 | 0.96 |
| No-4DSH | 29.05 | 0.96 | 33.71 | 0.97 | 31.38 | 0.97 |
| No-Time split | 28.89 | 0.96 | 32.86 | 0.97 | 30.25 | 0.97 |
| **Full** | **29.38** | **0.96** | **33.85** | **0.98** | **31.62** | **0.97** |

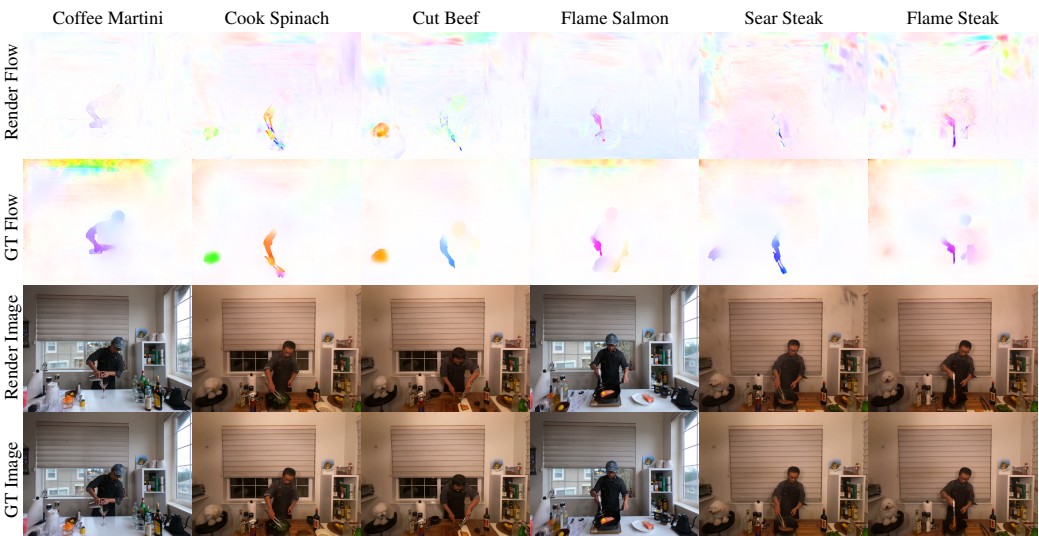

Figure 4: **Visualization of the emerged dynamics of our 4D Gaussian** The displayed views are selected from the test view of the Plenoptic Video dataset. The ground truth optical flows are extracted by VideoFlow (Shi et al., 2023) for reference.

a 3D displacement. To assess this scene motion capture ability, we conduct a thorough evaluation. For each Gaussian, we test the trajectory in space formed by the expectation of its conditional distribution $\mu_{xyz|t}$. Then, we project its 3D displacement between consecutive two frames to the image plane and render it using equation 6 as the estimated optical flow. In Figure 4, we select one frame from each scene in the Plenoptic Video dataset to exhibit the rendered optical flow. The result reveals that without explicit motion supervision or regularization, optimizing the rendering loss alone can lead to the emergence of coarse scene dynamics.

**More ablations** To examine whether modeling the spatiotemporal evolution of Gaussian's appearance is helpful, we ablate 4DSH in the second row of Table 3. Compared to the result of our default setting, we can find there is indeed a decline in the rendering quality. Moreover, when turning our attention to 4D spacetime, we realize that over-reconstruction may occur in more than just space. Thus, we allow the Gaussian to split in time by sampling new positions using complete 4D Gaussian as PDF. The last two rows in Table 3 verified the effectiveness of the densification in time.

## 5 CONCLUSION

We introduce a novel approach to represent dynamic scenes, aligning the rendering process with the imaging of such scenes. Our central idea involves redefining dynamic novel view synthesis by approximating the underlying spatio-temporal 4D volume of a dynamic scene using a collection of 4D Gaussians. Experimental results across diverse scenes highlight the remarkable superiority of our proposed representation, not only achieving state-of-the-art rendering quality but also delivering substantial speed improvement over existing alternatives. To the best of our knowledge, this work stands as the first ever method capable of real-time, high-fidelity video synthesis for complex, real-world dynamic scenes.

**Acknowledgments**   This work was supported in part by STI2030-Major Projects (Grant No. 2021ZD0200204), National Natural Science Foundation of China (Grant No. 62106050 and 62376060), Natural Science Foundation of Shanghai (Grant No. 22ZR1407500), USyd-Fudan BISA Flagship Research Program and and Lingang Laboratory (Grant No. LG-QS-202202-07).

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

# APPENDIX

## A    LIMITATIONS

While the first row of Figure 5 reveals that the 4D Gaussian can adeptly recover subsequent time steps exhibiting significant geometric deviations from the first frame via densification and optimization, using point clouds extracted from only the first frame, and achieves high fidelity in the foreground dynamic regions. Nevertheless, in the absence of initial points, our approach is difficult to capture distant background areas, even they are static. Although some techniques such as spherical initialization that we employed can mitigate this to an extent (for comparison, we turn off spherical initialization in the scene *coffee martini* in Figure 5, where it is evident that the exterior background was not successfully synthesized compared to the scene *flame salmon*), it does not suggest that we get the correct geometry - only a background map represented by a sphere of Gaussians is learned. These issues might constrain the convenience of our method in some scenes.

## B    PROOFS

In this chapter, we will prove the properties on which the main text relies when treating the unnormalized Gaussian defined in Eq. equation 1 as a special probability distribution.

First we will prove Gaussian conditional probability formula also holds in equation 5 and equation 6, i.e.

$$p(u, v, t) = p(t)p(u, v|t), \tag{12}$$

where

$$p(u, v, t) = (2\pi)^{-\frac{3}{2}} \det(\Sigma)^{-\frac{1}{2}} \mathcal{N}(u, v, t|\mu, \Sigma) \tag{13}$$

$$p(t) = (2\pi)^{-\frac{1}{2}} \det(\Sigma_t)^{-\frac{1}{2}} \mathcal{N}(t|\mu_t, \Sigma_t) \tag{14}$$

$$p(u, v|t) = (2\pi)^{-1} \det(\Sigma_{uv|t})^{-\frac{1}{2}} \mathcal{N}(u, v|\mu_{uv|t}, \Sigma_{uv|t}). \tag{15}$$

To prove equation 12, since Gaussian conditional probability formula holds for normalized version: $\mathcal{N}(u, v, t|\mu, \Sigma) = \mathcal{N}(t|\mu_t, \Sigma_t)\mathcal{N}(u, v|\mu_{uv|t}, \Sigma_{uv|t})$, we only need to prove

$$\det(\Sigma) = \det(\Sigma_t) \det(\Sigma_{uv|t}). \tag{16}$$

From Gaussian property, we know that $\Sigma_{uv|t} = \Sigma_{uv} - \Sigma_{uv,t}\Sigma_t^{-1}\Sigma_{t,uv}$. Then from the decomposition of $\Sigma$ by

$$\Sigma = \begin{bmatrix} \Sigma_{uv} & \Sigma_{uv,t} \\ \Sigma_{uv,t} & \Sigma_t \end{bmatrix} \tag{17}$$

$$= \begin{bmatrix} I & -\Sigma_{uv,t}\Sigma_t^{-1} \\ 0 & I \end{bmatrix} \begin{bmatrix} \Sigma_{uv} - \Sigma_{uv,t}\Sigma_t^{-1}\Sigma_{t,uv} & 0 \\ 0 & \Sigma_t \end{bmatrix} \begin{bmatrix} I & 0 \\ -\Sigma_t^{-1}\Sigma_{t,uv} & I \end{bmatrix}, \tag{18}$$

equation 16 holds immediately. The proof can be easily extended onto $p(x, y, z, t) = p(t)p(x, y, z|t)$.

## C   ADDITIONAL QUANTITATIVE RESULTS AND VISUALIZATIONS

In Table 4, we provide the PSNR breakdown on different scenes. Figure 5 shows more synthesis results at different timesteps for each scene in the Plenoptic dataset. The quanlitative results clearly show that our 4DGS is capable to faithfully capture the subtle movement of cookware. From the result in *Cut Roasted Beef*, we can find that even though we only use the point cloud extracted from the first frame as the initialization of the Gaussians, it is still able to fit the body after a large movement with high fidelity.

|  | Coffee Martini | Spinach | Cut Beef | Flame Salmon | Flame Steak | Sear Steak | Mean |
|---|---|---|---|---|---|---|---|
| HexPlane | - | 32.04 | 32.55 | 29.47 | 32.08 | 32.39 | 31.70 |
| K-Planes-explicit | 28.74 | 32.19 | 31.93 | 28.71 | 31.80 | 31.89 | 30.88 |
| K-Planes-hybrid | 29.99 | 32.60 | 31.82 | 30.44 | 32.38 | 32.52 | 31.63 |
| MixVoxels | 29.36 | 31.61 | 31.30 | 29.92 | 31.21 | 31.43 | 30.80 |
| NeRFPlayer | 31.53 | 30.56 | 29.353 | 31.65 | 31.93 | 29.12 | 30.69 |
| HyperReel | 28.37 | 32.30 | 32.922 | 28.26 | 32.20 | 32.57 | 31.10 |
| Ours | 28.33 | 32.93 | 33.85 | 29.38 | 34.03 | 33.51 | 32.01 |

Table 4: **Per-scenes results on the Plenoptic Video dataset.** we colored the Best, Second and Third results in each scene.

## D   RESULTS IN THE URBAN SCENES

Urban street bustling with numerous moving vehicles and pedestrians is one of the most common dynamic scenes in daily life. Reconstruction of such scenes has great value as it can provide rich data for the training of autonomous driving models and be used for offline perception.

To test the applicability of the proposed 4D Gaussian on the urban scenes, we selected several segments containing dynamic objects from the widely used Waymo Open Dataset. Each segment contains a sequence of calibrated images captured by five pinhole cameras and LiDAR point clouds which can not only provide an accurate initialization of 4D Gaussians but also be used for depth supervision. Following the previous work, we use images captured from three frontal cameras.

While the motion in urban scenes tends to be less intricate than that in typical indoor dynamic novel synthesis datasets, the sparse observation and the wide range pose different challenges. To mitigate the potential overfitting, we integrate sparse depth supervision sourced from LiDAR point clouds, given by the (inverse) L1 loss and we deactivate the temporal coefficient of 4DSH. Besides, we adopt a cube map as the background model to model the sky with infinite distance and penalize the inverse depth in the sky area. The sky mask is obtained using SegFormer (Xie et al., 2021).

In Figure 6, we provide the qualitative result of reconstruction. As can be seen, 4D Gaussian Splatting achieves high-fidelity rendering for both dynamic and static regions. More video results can be found in the supplementary material.

Furthermore, we present the novel view synthesis results in Figure 7. Following the common practice, we take out one frame from every ten frames as the test view. Unlike the previous approaches typically rely on the 3D bounding boxes and dynamic object segmentations, we provide a unified representation of both dynamic and static regions with the aforementioned modification.

## E   THE TEMPORAL CHARACTERISTIC OF 4D GAUSSIANS

If the 4D Gaussian has only local support in time, as the 3D Gaussian does in space, the number of 4D Gaussians may become very intractable as the video length increases. Fortunately, the anisotropic characteristic of Gaussian offers a prospect of avoiding this predicament. To further unleash the potential of this characteristic, we set the initial time scaling to half of the scene's duration as mentioned in Section 4.2.

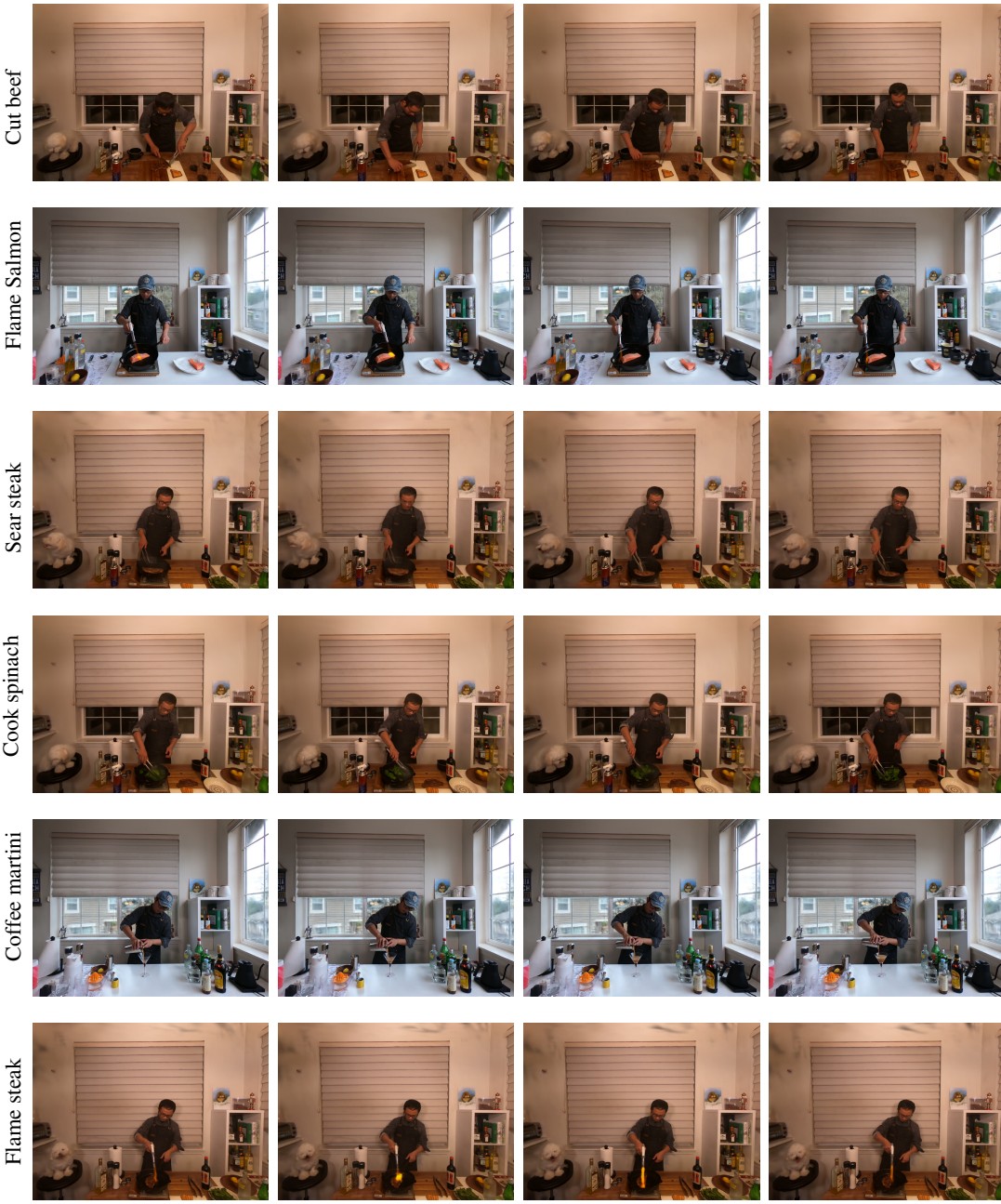

Figure 5: **View synthesis results at different times.**

In order to more intuitively comprehend the temporal distribution of the fitted 4D Gaussian, Figure 8 presents a visualization of mean and variance in the time dimension, by which the marginal distribution on $t$ of 4D Gaussians can be completely described.

It can be observed that these statistics naturally form a mask to delineate dynamic and static regions, where the background Gaussians have a large variance in the time dimension, which means they are able to cover a long time period. Actually, as shown in Figure 9, the background Gaussians are able to be active throughout the entire time span of the scene, which allows the total number of Gaussians to grow very restrained with the video length extending.

Moreover, considering that we filter the Gaussians according to the marginal probability $p_i(t)$ at a negligible time cost before the frustum culling, the number of Gaussians actually participated in the

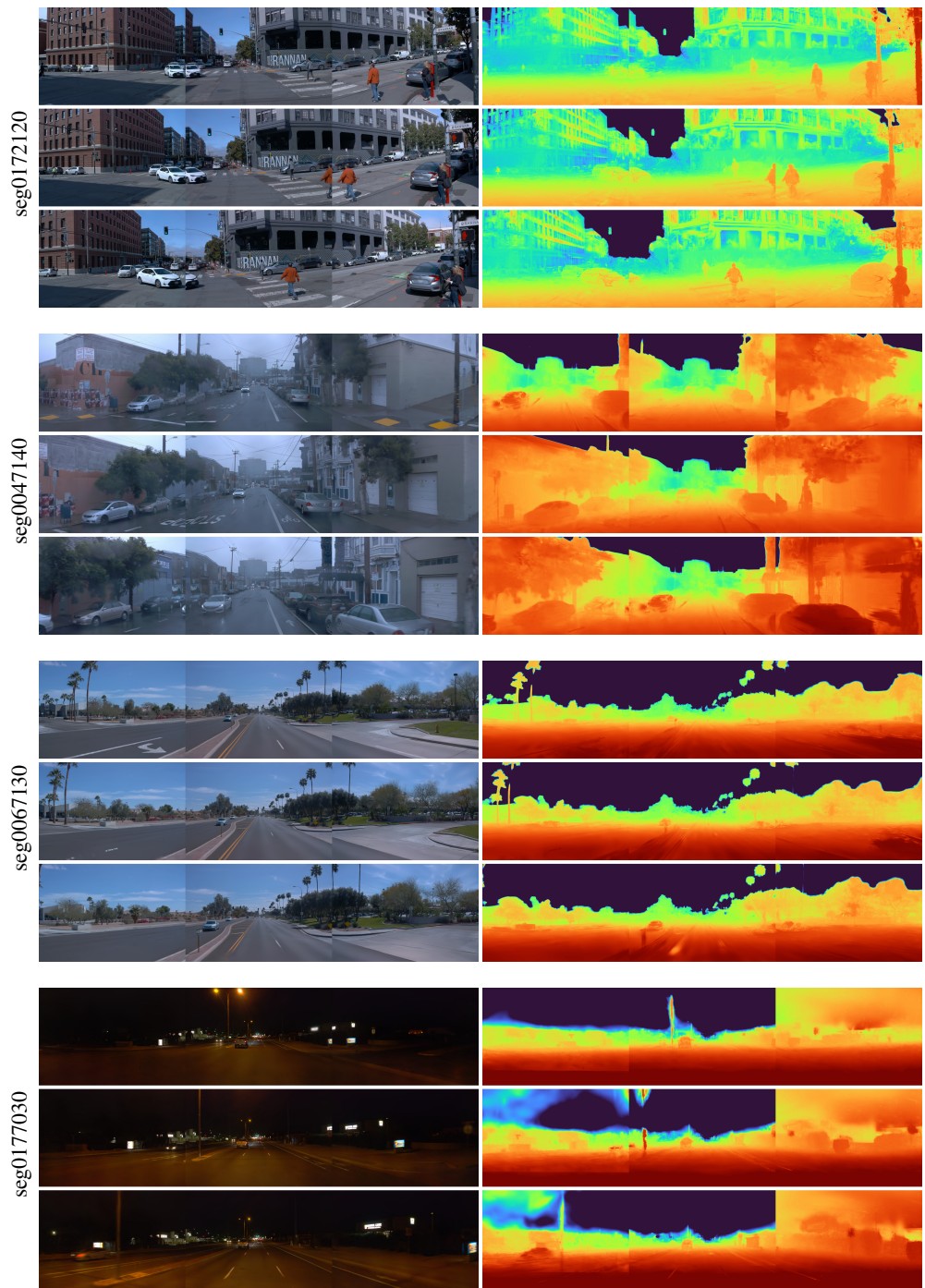

Figure 6: **Visualization of urban scene reconstructions.** **Left**: rendered RGB images. **Right**: rendered depth maps. Best viewed with zoom-in.

rendering of each frame is nearly constant, and thus the rendering speed tends to remain stable with the increase of the video length. This locality instead makes our approach friendly to long videos in terms of rendering speed.

In Figure 10, we directly show the total number of Gaussians and the number of Gaussians really involved in the rasterization for a given frame under different video lengths. As it can be seen, the total number of 4D Gaussians fitted on the video with hundreds of frames is not essentially larger

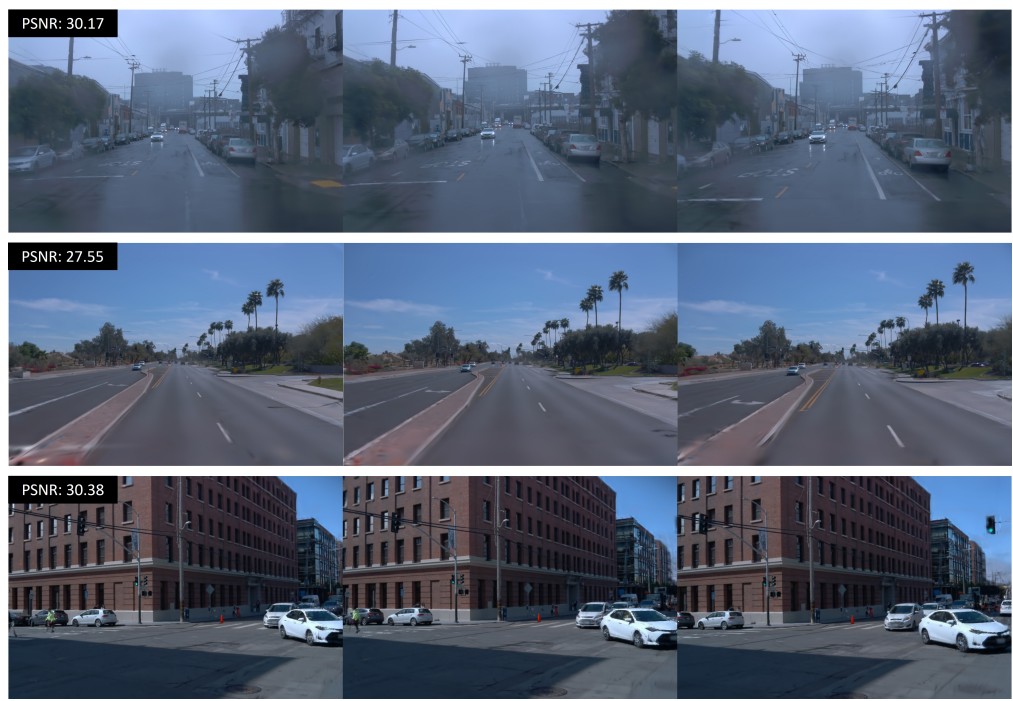

Figure 7: **Novel view synthesis in the urban scenes.**

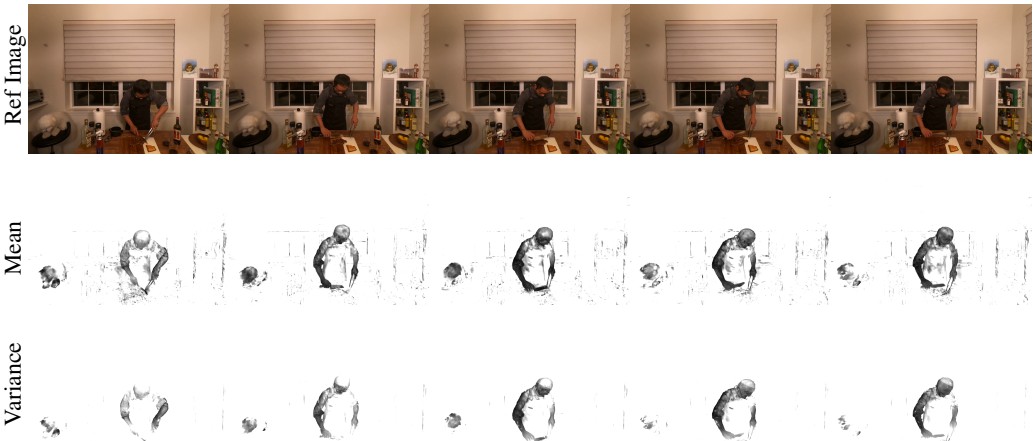

Figure 8: **The temporal distribution of the fitted 4D Gaussian on the *cut roasted beef*.** For better visualization, we show the distance between the mean and the rendered timestamp.

than that of 3D Gaussians fitted on a single frame and the average number of 4D Gaussians really used in rendering each frame is stable.

We compare the sliced 3D Gaussians of two variants (No-4DRot and Full) in Figure 11. It can be obviously observed that under the No-4DRot setting the rim of the wheel is not well reconstructed, and fewer Gaussians are engaged in rendering the displayed frames after filter, despite a larger total number of fitted Gaussians under this configuration. This indicates that the 4D Gaussian in the No-4DRot setting has less temporal variance, which impairs the capacity of motion fitting and exchanging information between successive frames, and brings more flicker and blur in the rendered videos.

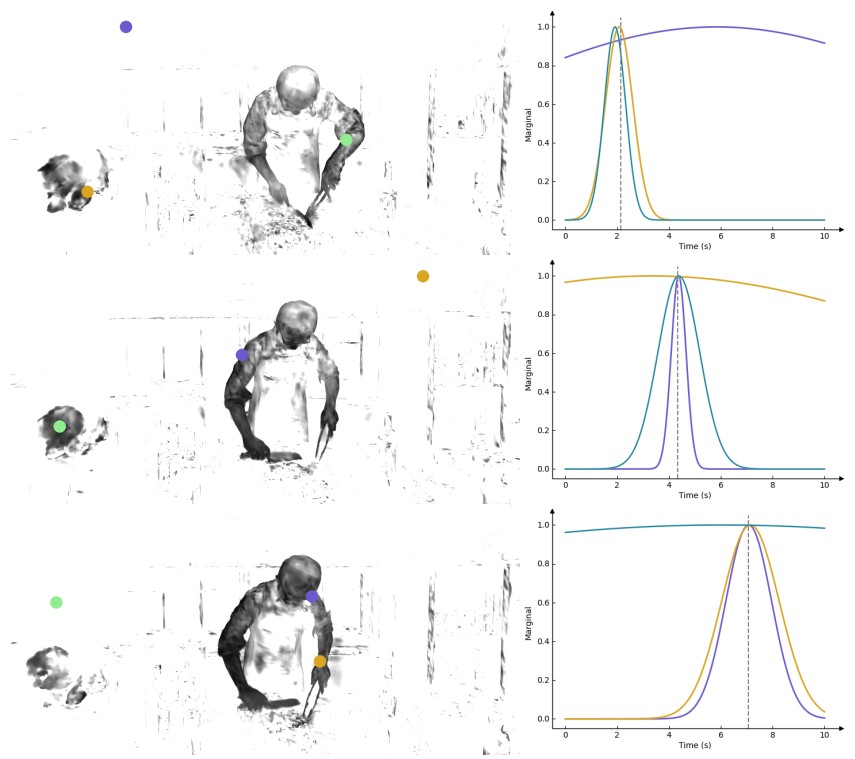

Figure 9: **The marginal distribution of the fitted 4D Gaussian in time dimension.**

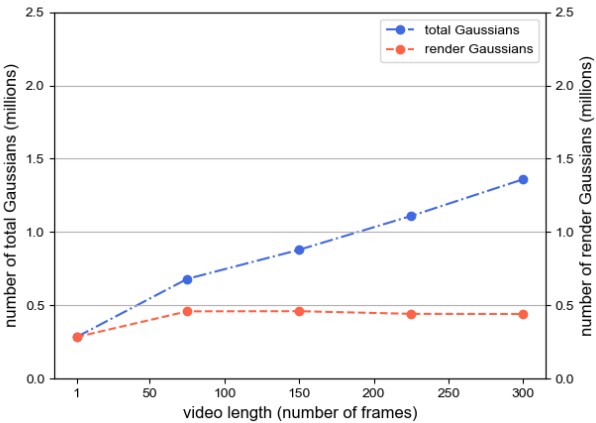

Figure 10: **The number of Gaussian at 10k iterations under different video lengths. The data at 1 frame denotes the number of 3D Gaussians fitted on the initial frame.**

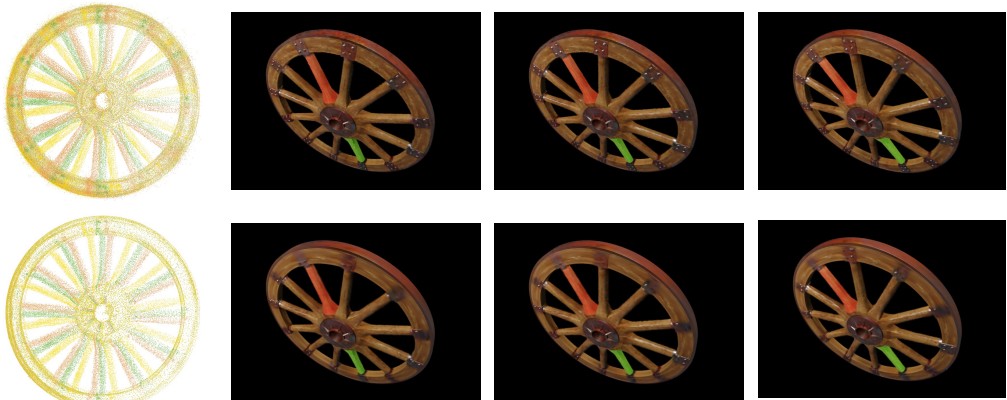

Figure 11: **Visualization of the time slices under Full setting (top) vs. No-4D Rot (bottom).** In the first column, we show the time slices of fitted 4D Gaussian under different settings. In the other columns, we present the rendered images.

