# OpenReview forum: "Real-time Photorealistic Dynamic Scene Representation and Rendering with 4D Gaussian Splatting"
_ICLR.cc/2024/Conference — ICLR 2024 poster_

### Official Review · Reviewer_8jaG · 2023-10-29

**Soundness:** 3 good
**Presentation:** 3 good
**Contribution:** 2 fair
**Rating:** 6
**Confidence:** 3

**Summary:**

This work is on dynamic scene representation and rendering.
The authors extend Gaussian Splatting to dynamic scenes by extending the 3D Gaussian formulation with $t$, along with 4D spherindrical harmonics.
Experiments are conducted on Plenoptic Video and D-NeRF dataset and demonstrate the effectiveness of the proposed method: more realistic and faster.
Analysis shows that some primitive underlying 3D movement can be captured.

**Strengths:**

1. The extension of 3D Gaussian for dynamic scenes. It allows using 3D Gaussian to render dynamic scenes realistically.
2. Overall the paper is clearly written and well explained in the motivation, design and implementation.
3. Superior results compared to very recent SOTA methods include K-Planes, HexPlanes. Higher realism and speed.

**Weaknesses:**

* Concerns Regarding Novelty:
   - The enhanced speed and realism seem to be primarily attributed to the power of Gaussian splatting, rather than the novel modules introduced.
   - The implementation of 4D spherical harmonics appears to offer only a marginal improvement, as evidenced by the mere +0.2 PSNR increase in Table 3.
   - The inclusion of time (t) in addition to spatial dimensions $(x,y,z)$ in the Gaussian model appears to be a straightforward extension, akin to extending TensoRF’s triplanes to K-planes.
* Modeling Concerns:
   - I have reservations about the authors’ approach of modeling the 4D Gaussian by treating space (xyz) and time (t) equally. This seems to suggest that each Gaussian fades in, peaked at $\mu_t$ and out, as indicated by the term $p$ in Equations 4 and 10. This is not grounded in physical reality, where Gaussians are expected to move through space rather than appearing and disappearing abruptly.
  - Consequently, I think this extension mainly increases the model’s capacity to incorporate time and I question whether the proposed approach can yield consistent results in more complex scenarios (e.g., higher moving speeds, sparse observations).


That being said, I think there are more effective ways to extend 3D Gaussian splatting for dynamics with a more accurate and physically-grounded representation. Nevertheless, I still consider this work to be above the bar in this field, providing some insights for the community.

**Questions:**

I know these are concurrent work (one even come after ICLR submission deadline). It would be admired if you can add in the final version about comparison to the following two works on extending Gaussian splatting for dynamic scenes.  I have no conflict of interests to these works.
1. 4D Gaussian Splatting for Real-Time Dynamic Scene Rendering. https://arxiv.org/pdf/2310.08528.pdf
2. Dynamic 3D Gaussians: Tracking by Persistent Dynamic View Synthesis  https://arxiv.org/pdf/2308.09713.pdf

Besides, it would be appreciated if you can apply the proposed method on any urban scenes (e.g. KITTI, Waymo dataset). These are the more challenging and common dynamic scenes in real world.

---

> ### Author Response · Authors · 2023-11-21
> **Response to Reviewer 8jaG**
>
> We thank the reviewer for the positive and detailed review as well as the suggestions for improvement. Our response to the reviewer’s comments is below:
>
> **Q1: The comparison with the concurrent work.**
>
> Very reasonable suggestion. We have updated the comparisons with the 4DGS [1] in the revised paper. For the Dynamic 3D Gaussian [2], we have discussed its differences in the related work section. In the discussion period, we additionally test its performance in the cut roasted beef scene using its officially released code. We present the comparison in the table below:
>
> | Method              | PSNR $\uparrow$  | SSIM $\uparrow$ | LPIPS $\downarrow$ |
> | :-----              | :---: | :---:  | :---: |
> | Dynamic 3D Gaussian [2] | 29.20 | 0.9551 | 0.08846 |
> | Ours | 33.85 | 0.9596 | 0.04057 |
>
> Note that the way described in the paper of Dynamic 3D Gaussian for obtaining the foreground/background mask cannot be simply applied to this scene, and its performance highly depends on these masks significantly. So we extract masks using an advanced video object segmentation model Xmem [3] with the masks for the first frame acquired via SAM [4]. In addition, we enable optimization of seg colors in the initial timestep and detach other parameters before rendering segmentation masks. Due to the above customization is not contained in the original Dynamic 3D Gaussian, we did not append this result in our revised paper.
>
> > [1] Wu, Guanjun, et al. 4d gaussian splatting for real-time dynamic scene rendering. *arXiv preprint*, 2023.
>
> > [2] Luiten, Jonathon, et al. Dynamic 3d gaussians: Tracking by persistent dynamic view synthesis. *3DV*, 2023.
>
> > [3] Cheng, Ho Kei, and Alexander G. Schwing. Xmem: Long-term video object segmentation with an atkinson-shiffrin memory model. *ECCV*, 2022.
>
> > [4] Kirillov, Alexander, et al. Segment anything. *ICCV*, 2023.
>
> **Q2: The application on the urban scenes.**
>
> Great thought. While this is beyond the scope of this work, we are delighted to share our explorations in this area. We test 4D Gaussians with minimal modification on selected dynamic sequences in the widely used Waymo Open Dataset. The modifications are summarized in the revised appendix, where we also provide some visualization of reconstruction and novel view synthesis results. The rendered videos can be found in the **supplementary material**. These results reveal that the proposed method can uniformly model both dynamic and static regions and achieve competitive reconstruction speed and quality without relying on any manually labeled dynamic mask. This indicates good potential of the proposed method.
>
> **Q3: Novelty concern.**
>
> In a sense, all work that extends static reconstruction methods to dynamic scenes can be viewed as the inclusion of time in addition to spatial dimension; how to achieve this is however non-trivial. Often such extensions will introduce new modules and optimizations, making the whole pipeline much more complex and less flexible. In this work, we implement a different extension by introducing an easily optimized and rendered parameterization of 4D Gaussian, and adapt the original rendering formulation in 3DGS. We believe that this extension is more flexible and inherits the advantages and spirits of the explicit modeling in 3D Gaussian Splatting, while it’s also fully interpretable and friendly to composition and editing. The 4DSH is designed as an integral part for appearance evolution over time. Indeed, the performance gain it brings is not impressive despite its obvious usefulness. Per our observation, this is due to limited cases of such time-dependent appearance involved within the evaluation video sequences. We will attempt to explore more extensive test data in the future work.
>
> **Q4: Modeling concern.**
>
> The issue raised by the reviewer 8jaG is common to splatting-based methods: we can draw an analogy in the 3D case, where many real-world 3D entities are composed of line segments with constant transparency, yet the 3D Gaussian also fades out beyond its $\mu_{xyz}$. However, this issue doesn’t significantly impede its expressive capability. The bullet times experiments in our supplementary material also suggest that this issue does not affect the temporal smoothness of 4D Gaussian. Furthermore, the unnormalized Gaussian function allows the 4D Gaussian to always have larger influence and does not fade out abruptly after $\mu_t$.
> For the issue of physically-grounded, it is better to clarify that  we are not aiming to model physical reality, and the 4D Gaussian should not be seen as corresponding to some physical entity. Actually, most of the successful methods for the task of novel view synthesis rather than simulation are not completely physically-grounded, although they may draw inspiration from some aspect of physical reality. Nevertheless, we believe that solving this issue properly might result in a new more advanced representation paradigm in the future research.

---

### Official Review · Reviewer_1bTN · 2023-10-30

**Soundness:** 3 good
**Presentation:** 3 good
**Contribution:** 3 good
**Rating:** 6
**Confidence:** 4

**Summary:**

This paper generalizes the 3D Gaussian splitting to 4D. The key contribution is a 4D Gaussian-based representation for dynamic rendering. Unlike intuitively modeling the 4D as canonical 3D + deformations, this paper inserts many 4D bubbles into 4D space-time and jointly assembles the dynamic scenes.

Interestingly, when using full rotational 4D Gaussians, to render a specific time, one just needs to slice the 4D distribution and then do standard 3D Gaussian EWA. The skew in 4D also leads to time-dependent center changing, which leads to local scene flow.

The proposed representation is tested with the dynamic rendering task on D-NeRF and DyNeRF datasets and shown to be effective. With the power of splitting, the proposed representation achieves high inference FPS.

**Strengths:**

- It’s interesting to use local 4D Gaussians to represent full 4D functions instead of using explicit flow plus 3D Gaussian.
- The introduction of full 4D covariance and the usage of the time-dependent harmonics of appearance is effective.
- The flow extracted from the time-dependent center sliced from 4D Gaussians makes this 4D function approximation more reasonable because it may locally capture correspondence, which introduces the smoothness into the 4D approximation.

**Weaknesses:**

- One key weakness the reviewer is curious about is the number of Gaussians in the proposed representation when the video grows longer. And how long does each 4D Gaussian cover in time? If the 4D Gaussians, like the 3D static one, only have local support in time, then the memory or storage consumption may be huge, preventing applying such representation to long videos or streaming.
- To help the readers understand the 4D Gaussians, one 2D/1D slice + time visualization may be helpful.
- The LPIPS is not reported in the table while most dynamic rendering does.
- Although the local flows as I write in the strengths are interesting, the reviewer is wondering how can these local correspondences generalize to global and long-term ones. It’s unclear whether such 4D mixture models can capture long-term tracking.

**Questions:**

Pelase see weakness

---

> ### Author Response · Authors · 2023-11-21
> **Response to Reviewer 1bTN**
>
> We thank the reviewer for the positive and detailed review as well as the suggestions for improvement. Our response to the reviewer’s comments is below:
>
> **Q1: The number of 4D Gaussians.**
>
> The reviewer’s insight on the locality is very discerning. Actually, unlike spatial scaling which is set to the mean of the distance to the closet three points, we initialized a very large scaling in the temporal dimension as mentioned in implementation details, which allows most background Gaussians to cover a long time period (around the scene’s duration), ensuring that the total number of the fitted 4D Gaussians does not significantly exceeds that of 3D Gaussians.
>
> As for the memory or storage consumption, we believe that there is no free lunch in this aspect. For any representation, with the increase of the video length, the storage requirements are unlikely to remain constant forever, because we can't store more information without any extra cost. Nevertheless, despite the possible storage consumption issues, the rendering speed of the proposed 4D Gaussian tends to be fixed as the video length increases. From this point of view, this characteristic of locality enables our proposed method to be more friendly to long videos than that of encoding information into implicit neural networks.
>
> **Q2: The visualization of time**
>
> Good suggestion. Since the temporal characteristics of the 4D Gaussian can be summarized by its variance and mean, we provide visualizations of them in Figure 8 of the revised appendix. Interestingly, these visualizations naturally form a mask of dynamic and static regions. This phenomenon not only shows the potential and the versatility of our approach, but also demonstrates that a big proportion of Gaussians are used for background modeling, with a long span over time. To demonstrate this more directly, we plot the marginal distribution of some representative Gaussians in Figure 9.
>
> **Q3: LPIPS**
>
> In the original submitted version, we only reported PSNR and SSIM following kplanes. Now we have updated LPIPS in the revised paper.
>
> **Q4: Long-term tracking**
>
> This is a great point and worth for deep investigation including both benchmark upgrade and algorithm advance. We consider this issue to be one of the most important aspects for extending our model. Considering that maintaining global correspondence may compromise the fitting capability and flexibility in complex dynamic scenes, we leave it for future work.

---

> > ### Comment · Reviewer_1bTN · 2023-11-21
> > **Keep Positive**
> >
> > Thanks for the author's response. After reading other reviews and the author's rebuttal, I currently tend to keep my positive recommendation of this paper.

---

### Official Review · Reviewer_XBJp · 2023-11-01

**Soundness:** 3 good
**Presentation:** 3 good
**Contribution:** 3 good
**Rating:** 8
**Confidence:** 5

**Summary:**

This submission introduces a 4D spatial-temporal representation (3D for space and 1D for time) based on Gaussian principles to model dynamic 3D scenes. This representation is a principled extension of the 3D Gaussian representation used in Gaussian Splatting for static scenes. With the 4D Gaussian representation, the states of 3D Gaussians at specific time instances become slices of the 4D function, enabling their rasterization into video frames for forward image rendering and scene reconstruction. The submission demonstrates that by modeling the interplay between space and time, implemented as a non-block-diagonal 4D covariance matrix, we can improve the modeling and reconstruction of dynamic scenes. The paper illustrates that with only the reconstruction loss and no additional regularizations, the 4D Gaussian-based representation achieves superior performance compared to state-of-the-art methods.

**Strengths:**

+The proposed 4D Gaussian based representation to model the spatial-temporal field is a very principled extension of the 3D GS method. As a result, the fitting process is robust - no need to introduce additional regularization terms or optimization strategy to make sure the optimization will converge.

+The scene reconstruction using the representation achieves SOTA performance with only reconstruction loss across all experiments.  No manually designed regularization term is needed. This shows the effectiveness of the 4D representation.

+The 4D densification and pruning operations in spacetime can be derived naturally by extending from 3D. As shown in the submission, those operations improve the performance on modeling scene motions

+The paper is well written with consistent and accurate notations, thus easy to follow and grasp.

**Weaknesses:**

- more discussion needed for key advantage of the proposed model:
Although in Sec 3.2 and Tab.3, the authors mentioned the benefit of making space and time dependent by allowing non-block diagonal 4D covariance matrix for modeling motions,  it is not easily seen why the performance increase (full version vs "No-4DRot"). The visualization of temporal slices of the fitted 4D Gaussian could be helpful to demonstrate how the corresponding 3D Gaussian across frames models motions

- more discussion on design choice needed:
The 4D SH function is also a function of time t, will it also explain the temporal appearance change (eg. motion)? if so, it is explaining the properties we did not intend it to (we use it for view-dependency), and it may interact with other optimized variables such as 4D rotation to explain the motion, will it lead to ambiguity and affect the performance of dynamic modeling ?

**Questions:**

1. The number of 4D Gaussians might be huge (millions in 3D GS paper), given that one more temporal dimension in addition to 3D space as in 3D GS. Can the author address that, since that might be one limitation of the Gaussian based representation?

2. Are the 4D means of the Gaussians also being optimized? If so, the 4D means can also introduce displacements to 3D Gaussians. In other words, both 4D mean offsets, and 4D rotation can lead to displacements. Will this introduce ambiguity to explain the scene flow field?

---

> ### Author Response · Authors · 2023-11-21
> **Response to Reviewer XBJp**
>
> We thank the reviewer for the positive feedback and constructive suggestions. Our response to the reviewer’s concerns is below:
>
> **Q1: The number of 4D Gaussians.**
>
> This is a great question regarding the scalability of our 4D Gaussian model. We further clarify the following facets:
>
> 1. The total number of 4D Gaussians is not essentially larger than that of 3D Gaussians, as each 4D Gaussian has a certain time span and there exists a high proportion of background Gaussians active throughout the whole time duration of the scene. For the typical scenes in the DyNeRF dataset, the number of 4D Gaussian fitted on the videos with 300 frames is on the same order (millions) as the number of points in the 3D Gaussian fitted only on the first frame.
>
> 2. Besides, since we filter the Gaussians according to the marginal of time with a negligible time cost before the frustum culling, the number of Gaussians actually participated in the rendering of each frame is nearly constant, and thus exhibiting stable rendering speed with the increase of the video length and the total number of Gaussians. The statistic of Gaussian’s number is provided in Figure 10 of the revised appendix.
>
>
> **Q2: The potential conflict in the jointly optimization of the 4D means and the rotation.**
>
> Great question. Yes, the 4D means are also optimized during training, but they are fixed during inference and their displacements are derived from the 4D rotations only. So there is no ambiguity during inference.
> Regarding the potential interference caused by the joint optimization of the means and rotation, we show in the following that their interaction will not introduce ambiguity of displacement. Consider a simplified linear example for illustration purposes: $y(t; x_0, v) = x_0 + vt$, where $x_0$ and $v$ are the trainable parameters. Given sufficient observations {$(y_i, t_i)$}$_{i=1}^{N}$, it can converge to the unique correct solution, despite that $x_0$ and $v$ also interact with each other during optimization. However, our model is slightly different. Specifically, our model should rather be abstracted as the following system: $y(t; x_0, v, t_0) = x_0 + vt_0 + vt$. Fortunately, if we do a simple substitution $w = x_0 + vt_0$, the above equation can be reformulated as $y(t; w, v) = w + vt$, allows for the unambiguous determination of $v$, ensuring that there is no ambiguity in the learning of displacement.
>
>
> **Q3: Discussion on the advantage compared to the naïve baseline.**
>
> We ascribe this performance gain to the motion modeling capabilities stemming from the proposed  4D rotation. As shown in Figure 4 of the original paper, the displacement derived from 4D rotation can roughly capture the underlying dynamics of the scene, which enables more efficient information sharing across frames and leads to more seamless transitions between them.
> We also provide more visualizations and discussions of temporal slices for two versions of the 4D Gaussian in Section E and Figure 11 of the revised appendix.
>
> **Q4: Discussion on the 4D SH.**
>
> Thanks. The possible ambiguity could be substantially mitigated by delaying the training of time-dependent coefficients. Actually, rather than inducing ambiguity, 4DSH’s benefits in motion learning are more critical. Because unlike the static scenes, there exists temporal evolution in the appearance of static objects in dynamic scenes. Without a particular design as our 4DSH to tackle this variable, such a phenomenon can only be fitted by undesired motion that shouldn’t exist. Moreover, our experience indicates that 4DSH brings a positive impact on performance in most cases.

---

> > ### Comment · Reviewer_XBJp · 2023-11-21
> >
> > Thanks for the additional discussion and content based on the review. My questions and concerns have been addressed and I appreciated the modification for the appendix to include more visualizations. I will keep the initial positive score.

---

### Meta-Review · Area_Chair_qzPF · 2023-12-10

**Metareview:**

The submission introduces a method that extends Gaussian splatting to dynamic scenes.  Reviewers liked the overall formulation and the strong results, though they also had concerns regarding the novelty and presentation. The authors did a good job during the rebuttal phase, after which all reviewers were supportive of its acceptance. The AC agrees.  The authors are encouraged to revise the submission to further incorporate the additional comments from the reviewers (some were posted after the author-review discussion period.)

**Justification For Why Not Higher Score:**

The proposed method is similar to a few concurrent works, which also extend the Gaussian splatting formulation to 4D.

**Justification For Why Not Lower Score:**

All reviewers are supportive of its acceptance due to the clean formulation and good results.

---

### Decision · Program_Chairs · 2024-01-16

Accept (poster)